# An Integrated Proteomic and Glycoproteomic Investigation Reveals Alterations in the *N*-Glycoproteomic Network Induced by 2-Deoxy-D-Glucose in Colorectal Cancer Cells

**DOI:** 10.3390/ijms23158251

**Published:** 2022-07-26

**Authors:** Cheng Ma, Hong-Yuan Tsai, Qi Zhang, Lakmini Senavirathna, Lian Li, Lih-Shen Chin, Ru Chen, Sheng Pan

**Affiliations:** 1The Brown Foundation Institute of Molecular Medicine, University of Texas Health Science Center at Houston, Houston, TX 77030, USA; herath.lakmini.senavirathna@uth.tmc.edu; 2Division of Gastroenterology and Hepatology, Department of Medicine, Baylor College of Medicine, Houston, TX 77030, USA; hong-yuan.tsai@bcm.edu (H.-Y.T.); ru.chen@bcm.edu (R.C.); 3Department of Pharmacology and Center for Neurodegenerative Disease, Emory University School of Medicine, Atlanta, GA 30322, USA; qi.zhang@emory.edu (Q.Z.); lli5@emory.edu (L.L.); lchin@emory.edu (L.-S.C.); 4Department of Integrative Biology and Pharmacology, University of Texas Health Science Center at Houston, Houston, TX 77030, USA

**Keywords:** 2-Deoxy-D-glucose, proteomics, glycoproteomics, mass spectrometry, colorectal cancer

## Abstract

As a well-known glycolysis inhibitor for anticancer treatment, 2-Deoxy-D-glucose (2DG) inhibits the growth and survival of cancer cells by interfering with the ATP produced by the metabolism of D-glucose. In addition, 2DG inhibits protein glycosylation in vivo by competing with D-mannose, leading to endoplasmic reticulum (ER) stress and unfolded protein responses in cancer cells. However, the molecular details underlying the impact of 2DG on protein glycosylation remain largely elusive. With an integrated approach to glycoproteomics and proteomics, we characterized the 2DG-induced alterations in *N*-glycosylation, as well as the cascading impacts on the whole proteome using the HT29 colorectal cancer cell line as a model system. More than 1700 site-specific glycoforms, represented by unique intact glycopeptides (IGPs), were identified. The treatment of 2DG had a broad effect on the *N*-glycoproteome, especially the high-mannose types. The glycosite occupancy of the high-mannose *N*-glycans decreased the most compared with the sialic acid and fucose-containing *N*-glycans. Many of the proteins with down-regulated high-mannose were implicated in functional networks related to response to topologically incorrect protein, integrin-mediated signaling, lysosomal transport, protein hydroxylation, vacuole, and protein *N*-glycosylation. The treatment of 2DG also functionally disrupted the global cellular proteome, evidenced by significant up-regulation of the proteins implicated in protein folding, endoplasmic reticulum, mitochondrial function, cellular respiration, oxidative phosphorylation, and translational termination. Taken together, these findings reveal the complex changes in protein glycosylation and expression underlying the various effects of 2DG on cancer cells, and may provide insightful clues to inform therapeutic development targeting protein glycosylation.

## 1. Introduction

Most cancer cells exhibit increased glycolysis and use this metabolic pathway for the generation of ATP as a main source of their energy supply, which is known as the Warburg effect and is considered to be one of the most fundamental metabolic alterations during malignant transformation [1]. Many efforts have been made to investigate the underlying mechanisms and the potential therapeutic implications of glycolysis inhibition for anti-cancer treatment [1,2]. 2-deoxyglucose (2DG) is a synthetic glucose analog in which the 2-hydroxyl group is replaced by hydrogen. Because of the high structural similarity between 2DG and glucose, 2DG can be readily taken up by glucose transporters and works as a competitive inhibitor of glycolysis to induce ATP deficiency, cell cycle arrest, inhibition of cell growth, and even cell death [3,4]. ATP deficiency is the main consequence of the inhibition of glycolysis, which alters the ATP/AMP ratio and activates AMP-activated protein kinase (AMPK) [5].

In addition to the inhibition of glycolysis, it has been found that 2DG modulates protein glycosylation through competing with D-mannose [6,7,8]. In Eukaryote, all *N*-linked glycan structures have a common core pentasaccharide (Man_3_GlcNAc_2_). The initial *N*-glycan structure consists of 14 sugar residues (Glc_3_Man_9_GlcNAc_2_) that are synthesized in the endoplasmic reticulum (ER) as a branched structure on a lipid-linked oligosaccharide (LLO) (Glc_3_Man_9_GlcNAc_2_-P-P-dolichol). Through a co-translational process, Glc_3_Man_9_GlcNAc_2_ is assembled from LLO to the Asn-X-Ser/Thr consensus sequence (where X is any amino acid except P) in proteins, and is then processed in the Golgi apparatus by various glycosyltransferases into high-mannose, complex, or hybrid structures [9]. The presence of glycans on a protein is believed to fine-tune the function of the protein, and their aberrance or absence could alter or even completely abolish the protein function [10,11,12].

By modulating protein glycosylation, 2DG can lead to ER stress and an unfolded protein response (UPR) in cells, which may be followed by cell dysfunction and death [13,14]. Elimination of the hydroxyl group at C-2 in the D-mannose molecule leads to the same 2DG compound; thus, 2DG also interferes with the metabolism of D-mannose, including disruption of the mannose-related metabolic pathways and competition with D-mannose in protein glycosylation [6,7,15,16]. 2DG incorporation into GDP-2DG competes with GDP-mannose for mannosyltransferases for synthesizing *N*-linked glycans on LLOs, resulting in partially glycosylated or non-glycosylated proteins [17]. Consequently, 2DG modified glycoproteins cause protein misfolding and activation of UPR to maintain cell viability or initiate cell death programs [18].

Currently, while metabolomic and functional proteomic studies have extensively explored the effects of 2DG on cells [19,20,21,22,23], a knowledge gap still exists in understanding the molecular details of the impact of 2DG on protein glycosylation. This is in part due to the challenges in analyzing intact *N*-glycopeptides because of the complex linkage among oligosaccharides and various glycan isomers. With the emerging approach of intact glycopeptide (IGP) analysis [24,25,26], herein, we report an in-depth analysis on the 2DG-induced *N*-glycosylation changes in HT29 human colorectal adenocarcinoma cells using an integrated approach with proteomics, glycoproteomics, and systems biology. The results revealed the effect of 2DG, as an anti-cancer agent, on the *N*-glycoproteome and associated functional networks, and might provide insightful clues to inform therapeutic development targeting protein glycosylation.

## 2. Results

### 2.1. Proteomic and N-Glycoproteomic Analyses of the 2DG-Treated HT29 Cells

A label-free proteomic analysis was applied to reveal the proteome alterations in HT29 colorectal cancer cells treated with 5 mM 2DG, using the untreated cells as a control. Three independent replicate experiments were performed to enhance the reproducibility and robustness of the method in the peptide and protein analysis. In total, 4257 unique proteins or protein groups were identified. Differentially expressed proteins associated with 2DG treatment were defined using a threshold of ±2-fold change of protein abundance in the 2DG treated HT29 over the control, with a Student’s *t*-test *p*-value < 0.05. A total of 325 up-regulated proteins and 94 down-regulated proteins were identified (Figure 1A and Appendix A). A group of 2DG-influenced proteins are shown in Figure 1B, including UGGT1, UGT1A10, ERP29, P53, TP53BP1, and GDI8. It has been reported that the overexpression of P53 can enhance cytotoxicity in 2DG-treated human tumor cells via oxidative stress. The p53-binding protein 1 (T53BP1) is a protein considered to be a transcriptional coactivator of the p53 tumor suppressor. ERP29 is a chaperone-binding protein involved in protein folding within the ER. UGGT1 is an important enzyme involved in glycoprotein reglucosylation and for providing quality control for protein transport out of the ER. DAP3, a ribosomal protein, is involved in mediating interferon-gamma-treated cell death. GID8 is the core component of the CTLH E3 ubiquitin−protein ligase complex that selectively accepts ubiquitin from UBE2H and mediates ubiquitination and subsequent proteasomal degradation of the transcription factor HBP1. These results suggest that 2DG have stimulatory effects on tumor cells in various biological processes.

The intact *N*-glycopetide analysis resulted in a total of 1738 IGPs identified in the HT29 cell lines, including 672 IGPs that were detected in all biological repeats under 2DG-treated or untreated HT29 cell lines (Appendix A). Figure 1C shows the hierarchical clustering of 114 IGPs (35 up-regulation and 79 down-regulation) that had significant alterations associated with 2DG treatment (≥2 or ≤0.5 in fold change, *p*-value < 0.05), including IGPs with *N*-glycans for the high-mannose, complex, and hybrid. These altered IGPs were identified from 77 proteins. Further analysis indicated that majority of these dysregulated IGPs, i.e., 82 IGPs (17 up-regulation and 65 down-regulation) from 61 proteins, were high-mannose containing *N*-glycopeptides (Figure 1D). The top 10 proteins with the most down-regulated *N*-glycosites are presented in Figure 1E. Hypoxia up-regulated protein 1 is the most down-regulated glycoprotein, with six down-regulated *N*-glycosites. This protein has a pivotal role in the cytoprotective cellular mechanisms triggered by oxygen deprivation and may play a role as a molecular chaperone in protein folding. Figure 1F shows the decrease in *N*-glycosylation occupancy at specific glycosites in EGFR (asn362), STT3B (asn627), UGGT1 (asn269), STTA1 (asn544), LAMP1 (asn261), and CD63 (asn130).

### 2.2. Functional Clustering of Dysregulated Proteins Induced by 2DG

The outcomes of the GO enrichment analysis of the dysregulated proteins associated with 2DG treatment are illustrated in Figure 2A (up-regulated proteins) and Appendix A (down-regulated proteins). The enriched biological processes by up-regulated proteins include protein folding, responding to ER stress, mitochondrial gene expression, cellular respiration, oxidative phosphorylation, translational termination, etc. The enriched GO-term of the cellular components demonstrated that 2DG treatments significantly influenced the proteins in the mitochondrial matrix, mitochondrial inner membrane, protein-containing complex, and ER lumen. The treatment of 2DG could stimulate molecular functions in the isomerase activity, ubiquitin-like protein ligase binding, electron transfer activity, and ligase activity, etc. For the down-regulated proteins, the only significantly enriched GO-term was molecular function (*p* < 0.05), which includes translation regulator activity, ligase activity, helicase activity, aminoacyl-tRNA ligase activity, etc. (Appendix A).

To interrogate the potential biological complexities in which a gene may belong to multiple annotation categories, we applied the cnetplot function [27] of the clusterProfiler package to depict the network linkages of the up-regulated proteins associated with the five top-ranked biological processes. Nine proteins, including UGGT1, ERP29, WFS1, DNAJC3, PDIA3, P4HB, PDIA4, and VCP, were annotated to two biological process terms—protein folding and response to ER stress (Figure 2B). Similarly, 11 proteins, including MRPS28, MRPL2, MRPL24, MRPL50, MRPL47, DAP3, MRPS35, MRPS15, MRPL11, MRPL11, and MRPL49, were annotated to three mitochondria related biological processes, including mitochondrial translation, mitochondrial translational elongation, and mitochondrial gene expression (Figure 2C). In addition, we applied a chord plot to visualize the association of the biological process GO-term categories with the relevant up-regulated proteins. We found that a total of 68 up-regulated proteins were involved in the eight top-ranked biological processes, including mitochondrial translational elongation, mitochondrial gene expression, protein folding, response to ER stress, and oxidative phosphorylation, etc. (Figure 2D).

### 2.3. Altered N-Glycoforms with High-Mannose, Hybrid, and Complex Types

To better understand the effects of 2DG on cellular *N*-glycome, we first grouped 1560 IGPs into three categories based on the general types of *N*-glycans, including high-mannose, complex, and hybrid. The results indicate that 2DG had a significant inhibition on intracellular protein *N*-glycosylation, especially for the high-mannose types (Figure 3A). The impacts of 2DG on complex and hybrid type of *N*-glycans were also observed, but to a considerably lesser extent. Further analysis of the glycan composition indicated that the abundance of high-mannose glycans had the most decreases compared with sialic acid and fucose-containing *N*-glycans (Appendix A). Subsequently, we classified the 1560 IGPs into 42 groups according to the *N*-glycan composition, and found that the decrease in the abundance of IGPs with high-mannsoe compositions were mainly associated with GlcNAc(2)Man(6), GlcNAc(2)Man(7), GlcNAc(2)Man(8), GlcNAc(2)Man(9), and GlcNAc(2)Man(10) (Figure 3B). Three MS2 spectra of intact *N*-glycopeptides, representing glycopeptides with high-mannose, hybrid, and complex glycan types, are illustrated in Figure 3C–E.

Among the proteins with down-regulated *N*-glycosylation induced by 2DG, several were notable because of their profound implications in cancer, such as galectin 3-binding protein (LGALS3BP), carcinoembryonic antigen-related cell adhesion molecule 1 (CEACAM1), and epidermal growth factor receptor (EGFR). LGALS3BP, a multifunctional glycoprotein involved in immunity and cancer, has seven consensus *N*-glycosites, and has been found heavily glycosylated in the tumor tissue and blood from pancreatic cancer patients [28,29]. In this study, we identified various glycoforms at four *N*-glycosites (Asn69, Asn125, Asn398, and Asn551) in this protein (Figure 4A). All of the glycoforms identified at the Asn69 and Asn125 sites only contained high-mannose *N*-glycans, and showed a significant decrease or even complete inhibition in glycosylation due to 2DG treatment (Figure 4B,C). The glycoforms at Asn398 and Asn551 included not only high-mannose, but also complex and hybrid *N*-glycans, and had a similar influence from 2DG treatment, in particular mannosylation (Figure 4D,E). CEACAM1 was reported to be up-regulated in metastatic colon cancer, with a suggested bimodal role in CRC progression. Herein, we identified a group of glycoforms associated with 4 N-glycosites in CEACAM1. Notably, two glycan compositions, including HexNAc(2)Hex(8) and HexNAc(4)Hex(5)Fuc(1)NeuAc(1) at Asn363, were identified only in the untreated HT29 cells, suggesting a significant inhibition of these glycoforms by 2DG. EGFR is a highly glycosylated transmembrane receptor tyrosine kinase with 11 potential consensus *N*-glycosites [30]. We identified Asn352 as a high-mannose *N*-glycosite with three glycan compositions, including HexNAc(2)Hex(6), HexNAc(2)Hex(7), and HexNAc(2)Hex(8). The percentages of glycosite occupancy of these glycoforms were substantially decreased after 2DG treatment.

### 2.4. Biological Theme Comparison among Different N-Glycosylation Types

It has been reported that a lower dose of 2DG (1 mM) induces a strong increase in mannose incorporation into cellular glycoproteins, whereas higher 2DG concentrations (10 and 20 mM) lead to a significant decrease in glycoprotein mannosylation in cancer cells. In this study, we found that the occupancy of high-mannose was significantly decreased with the treatment of 5 mM 2DG, whereas the impacts on sialic acid- and fucose-containing glycoforms were less. The GO-enrichment analysis comparing the three main *N*-glycan types, namely high-mannose (2HexNAc), hybrid (3HexNAc), and complex-type/bisecting (with more than 3HexNAc), confirmed that 2DG had a major impact on high-mannose compared with hybrid and complex/bisecting *N*-glycans. The enriched biological processes associated with the down-regulation of the high-mannose-type included the integrin-mediated signaling pathway, lysosomal transport, protein targeting to lysosome, protein hydroxylation, and protein *N*-glycosylation via asparagine, etc. (Figure 5A). The corresponding enriched molecular function included virus receptor activity, exogenous protein binding, integrin binding, transferase activity, transferring hexosyl group, retromer complex binding, and collagen binding, etc. (Figure 5B).

To reveal the potential biological complexity of the multiple subtypes of the altered *N*-glycoproteins that belong to different biological processes and molecular functions, we applied the cnetplot function [27,31] to extract the complex associations. Functional profiles of *N*-glycoproteins clusters revealed that 13 down-regulated high-mannose containing glycoproteins were involved in three enriched biological processes, namely response to topologically incorrect protein, protein hydroxylation, and protein N-linked glycosylation via asparagine (Figure 5C). Among these 13 proteins, TOR1B is a molecular chaperone. These biological processes were linked through three protein nodes: ERO1A, STT3B, and UGGT1. In addition, the altered *N*-glycoproteins containing hybrid and/or high-mannose *N*-glycans are jointly involved in the biological processes of lysomal transport, protein targeting to the lysosome, and protein targeting to vacuole. On the other hand, glycoproteins containing complex and/or high mannose *N*-glycans are jointly involved in the integrin−mediate signaling pathway (Figure 5C). Subsequently, we applied an enrichment map to organize the enriched terms (top 20 categories) into a network with edges connecting or overlapping the altered subtype *N*-glycoprotein sets (Figure 5D). Interestingly, a network consisting of four biological processes (response to unfolded protein, response to topologically incorrect protein, peptidyl-asparagine modification, and protein *N*-linked glycosylation) only included the altered *N*-glycoproteins with high-mannose glycans.

### 2.5. Interactive Influences of 2DG on Global Proteome and N-Glycoproteome

We compared the results between the glycoproteomic and proteomic analyses to assess whether the dysregulation of *N*-glycosylation induced by 2DG was correlated with the changes in protein abundance. Through the Venn diagram analysis, we found that there was little overlap between the glycosylation-dysregulated *N*-glycoproteins and the proteins with a differential expression (Appendix A). This observation suggests that most of the glycosylation changes detected in intact *N*-glycopeptides were not related to their core protein expression, and were probably caused by altered glycosylation occupancy rather than changes in the protein abundance. On the other hand, the changes in the *N*-glycan biosynthesis-related enzymes, notably glycosyltransferases (GTs), reflected an interactive interplay linking the alterations in *N*-glycosylation with the functional changes associated with protein expression. In eukaryotes, GTs are located in the ER or Golgi apparatus, and are responsible for the assembly, processing, and turnover of glycans. The biosynthesis of glycans is primarily determined by the GTs that assemble monosaccharide moieties into linear and branched glycan chains. At the proteomic level, we identified eight GTs (COLGALT1, DPM3, OGT, OSTC, RPN1, UGGT1, and UGT1A10), which were all located in the ER and were up-regulated after 2DG treatment. At the glycoproteomic level, we identified fiveGTs (RPN1, UGGT1, PLOD3, STT3A, and STT3B), of which four GTs (RPN1, UGGT1, PLOD3, and STT3A) showed a decrease in the *N*-glycosylation level.

## 3. Discussion

As a glycolysis inhibitor, 2DG mainly competes with glucose and inhibits glucose transport. Once inside the cell, 2DG is first phosphorylated by hexokinase II to 2-deoxy-d-glucose-6-phosphate (2DG-6-P). However, unlike glucose-6-P, 2DG-6-P cannot be further metabolized by phosphoglucose isomerase (PGI). Our proteomic results revealed an increase in the abundance of hexokinase II in HT29 cells after the treatment of 5 mM 2DG, implying a possible feedback mechanism due to the accumulation of 2DG-6-P within the cells. Additionally, facilitative glucose transporter member 1, which is responsible for constitutive or basal glucose uptake [32,33,34,35], was up-regulated in the 2DG-treated HT29 cells. As reported previously, oxygen deficiency led to a higher expression of glucose transporters and glycolytic enzymes, which could increase 2DG uptake in cancer cells compared with normal cells in an aerobic environment [36]. Thus, the excess 2DG might enter the cell through the glucose transporters and further stimulate its expression.

It is known that the most important consequence of glycolysis inhibition is ATP deficiency, which alters the ATP/AMP ratio and further activates AMP-activated protein kinase (AMPK) [5]. AMPK is regarded as the major energy-sensing kinase that activates a whole variety of catabolic processes in multicellular organisms, such as glucose uptake and metabolism. In addition, AMPK is a heterotrimeric protein complex that is formed by α, β, and γ subunits, each of which undertakes a specific role in both the stability and activity of AMPK [37,38]. Our proteomic results indicated that AMP-activated protein kinase catalytic α and γ subunits were increased in HT29 cell after 2DG treatment. Activated AMPK could promote autophagy, a degenerative mechanism present in every living cell, by phosphorylating autophagy-related proteins [39]. Additionally, a study has shown that 2DG-induced ER stress could stimulate autophagy [16]. In this study, the GO-term analyses did not indicate a significant enrichment of the biological processes associated with cellular autophagy. It was possible that the dose of 5 mM 2DG was not sufficient to cause an observable autophagy activation in the HT29 cell line. Another noteworthy finding of this study is that up-regulated proteins associated with 2DG treatment were highly enriched in the biological processes of mitochondrial gene expression, mitochondrial translation, and mitochondrial translation extension. The mitochondria have been shown to play a key role in tumorigenesis. In addition to having basic bioenergetic functions, mitochondrial metabolism provides the appropriate building blocks for tumor anabolism, controls redox homeostasis, and coordinates cell death [40]. Our results demonstrate that 2DG affected the biological processes of the mitochondria in the HT29 cell line, but more detailed studies are needed to elucidate how 2DG regulates mitochondrial biological processes in cancer cells.

In the glycoproteomic analysis, our findings uncovered a significant impact of 2DG on high-mannose *N*-glycans compared with the hybrid and complex types. The enriched GO-term analysis demonstrated that 2DG treatments resulted in protein misfolding and ER stress. In eukaryotic cells, carbohydrates are added primarily to unfolded proteins in the ER, and the cells use glycosylation to promote and regulate protein folding and quality control [41,42]. 2DG can compete with D-mannose to participate in glycan synthesis, ultimately leading to the misfolded glycoprotein in vivo [6,7,8]. As a consequence, these misfolded glycoproteins lead to ER stress. The ER degradation-enhancing α-mannosidase-like protein (EDEM) family proteins play a key role in the discrimination between proteins undergoing a folding process and misfolded glycoproteins for ER-associated protein degradation (ERAD) [43]. The schematic of 2DG implications on tumor cells is illustrated in Figure 6. Evidently, UDP-glucose glycoprotein glucosyltransferase 1 (UGGT1), and ER resident protein 29 (ERP29), which are involved in protein folding and ER stress, were down-regulated in the HT29 cells induced by 2DG. UGGT1 is a well-documented enzyme, which can recognize glycoproteins with minor folding defects and reglucosylate single *N*-glycans near the misfolded part of the protein, thus acting as a “sensor” to provide quality control for protein folding in the ER [44]. ERP29 is a chaperone protein, which plays an important role in the processing of secretory proteins within the ER, possibly by participating in the folding of proteins in the ER [45]. Our analyses suggest that the 2DG-incorporated *N*-glycoproteins are likely to be recognized by these critical proteins for protein refolding or degradation. It is noteworthy that while the expression of UGGT1 was increased at a protein level, the percentage of high-mannose glycan occupancy at Asn269 was decreased. Altered glycosylation at Asn269 might have a negative impact on the enzyme activity of UGGT1C, diminishing its role in guaranteeing a correct protein folding and thus enhancing ER stress.

Currently, 104 GTs families are recognized in the carbohydrate-active enzymes database (CAZy), 44 of which are represented in humans [12]. *N*-Glycans synthesis by GTs is processed in the ER and the Golgi. Interestingly, all of the altered GTs identified in our proteomic and glycoproteomic analyses were located in the ER. Some of these GTs, including dolichyl-diphospho oligosaccharide-protein glycosyltransferase subunits STT3A and STT3B, catalytic subunits of the oligosaccharyl transferase (OST) complex, catalyze the initial transfer of Glc_3_Man_9_GlcNAc_2_ from the lipid carrier dolichol-pyrophosphate to an asparagine residue, which is the first step in the protein *N*-glycosylation synthesis [46]. The 2DG-induced decrease in *N*-glycosylation of some GTs, such as the decreased high mannose occupancy of Asn544 of STT3A and Asn627 of STT3B, might have an impact on their biological activity and/or substrate selection in the ER, which in turn may affect their function as glycosyltransferases. As a consequence, the increase in the protein expression of some GTs might indicate a possible feedback mechanism in order to maintain homeostasis in protein glycosylation by resisting the influence of 2DG. Their alterations might contribute to ER stress and the related cellular processes by affecting *N*-glycosylation.

In a previous study, we demonstrated that by using a glutamine analog (6-diazo-5-oxo-L-norleucine, DON) to disrupt the glutamine metabolic pathways, including the hexosamine biosynthesis pathway (HBP) for synthesizing glycan building blocks, we could sensitize chemoresistant pancreatic cancer cells [47]. Similarly, as an analog of glucose, 2DG could also participate in the HBP route and impact glycan biogenesis. Our glycoproteomic results indicated that 2DG led to the changes in *N*-glycosylation of some well-known cancer-associated glycoproteins, including LGALS3BP, CEACAM1, and EGFR. The down-regulation of *N*-glycosylation in these proteins might impact their functions and the associated networks involved in cancer progression, immune response, and drug resistance.

## 4. Materials and Methods

### 4.1. Cell Culture and Treatment

The HT29 cell line was maintained in Dulbecco’s Modified Eagle’s Medium supplemented with 0.1 mM sodium pyruvate, 10% fetal bovine serum, penicillin (100 U/mL), and streptomycin (100 mg/mL). The cells were cultured in a humidified atmosphere at 37 °C in an incubator with 5% CO_2_. The cells were treated with 5 mM 2DG (Sigma) for 24 h. After incubation, the cells were collected and lysed in a RIPA buffer. The protein concentration was measured with the BCA assay (OD wavelength) using a Nanodrop UV-spectrometry (Thermo Scientific, Waltham, MA, USA).

### 4.2. Protein Purification and Trypsin Digestion

The cell lysate was collected and homogenized, and the cell debris were pelleted by centrifuging the lysate at 13,000× *g* for 15 min. The supernatant was collected and processed for both proteomic (50 µg) and glycoproteomic (200 µg) analysis, as previously described [48]. Samples were reduced with 10 mM DL-Dithiothreitol (DTT) at 50 °C for 1 h, and alkylated with 25 mM iodoacetamide at room temperature for 30 min in the dark. The proteins were precipitated by adding one fourth volume of 100% *w*/*v* tricholoracetic acid. The samples were incubated on ice for 10 min and centrifuged at 16,000× *g* for 10 min at 4 °C. The precipitate was washed with ice-cold acetone twice and air dried. The precipitate was suspended in 50 mM ammonium bicarbonate. Bovine thyroglobulin (*TG*) was spiked in the samples with a final concentration of 0.5% (µg/µg total protein) as an internal reference standard for the glycoproteomic analysis. The proteins were digested with Trypsin (1:30) in a two-step process. In the first step, half the amount of the enzyme was added and the samples were vortexed once each for 30 min for 2 h at 37 °C. Then, the other half of the enzyme was added to the samples and the samples were incubated at 37 °C overnight. After enzymatic digestion, the reaction was stopped by adding 1 volume of 1% formic acid. The final concentration of the peptide mixture was determined with a BCA assay using nanodrop UV-spectrometry (Thermo Scientific) using an extinction coefficient of 1.1 for 0.1% (g/L) solution at 280 nm. For the proteomic analysis, the samples were desalted using a self-packed C18 ZipTip microcolumn and were dried at room temperature in a speed vacuum concentrator. The samples were stored at −20 °C for further analysis.

### 4.3. Enrichment of Intact N-Glycopeptides

For the glycoproteomic analysis, 200 µg of digested sample was used to enrich the glycopeptides using a SOLA AX Cartridge (ThermoFisher Scientific, Waltham, MA, USA). The AX cartridge was conditioned with 1 mL acetonitrile (ACN) and 1 mL of 100 mM triethylammonium acetate (pH 7.0), and further equilibrated with 1 mL washing buffer (95% ACN and 1% TFA in H_2_O). The sample was reconstituted in 60 µL (or less) loading buffer (50% ACN and 0.1% TFA in H_2_O) and loaded onto the AX cartridge. The loading process was repeated twice. The cartridge was washed with 3 × 1 mL washing buffer to remove the non-glycosylated peptides. The glycopeptide fraction was eluted with 2 × 0.6 mL elution buffer (50% ACN and 0.1% TFA in water). The eluted glycopeptides were collected and vacuum dried. The samples were stored at −20 °C for further analysis.

### 4.4. LC–MS/MS Analysis

The LC–MS/MS analysis was performed on a Q-Exactive HF-X mass spectrometer (ThermoScientific) equipped with a nano-spray source interfaced with a *nano*LC UltiMate 3000 high-performance liquid chromatography system (Thermo Scientific). A self-packed C18 column (100 µm × 30 cm; particle size, 3 μm Reprosil-Pur C18-AQ beads) was used for the separation. Separation was achieved with a linear gradient from 3% to 40% solvent B for 120 min at a flow rate of 350 nL/min (mobile phase A, 99.9% H_2_O, 0.1% FA; mobile phase B, 80% ACN, 20% H_2_O, 0.1% FA). The Q-Exactive HF-X mass spectrometer was operated in the data-dependent acquisition (DDA) mode. The survey scan was acquired with 60,000 resolution from 400–1600 *m*/*z* with an AGC target of 1 × 10^6^ and a max injection time of 22 ms. The precursors were isolated in the quadrupole within an isolation window of 1.6 *m*/*z*. The top 25 monoisotopic masses were selected with a minimum intensity threshold of 5 × 10^3^ for fragmentation using higher energy collisional dissociation (HCD).

The analysis of the intact glycopeptides was performed on an Orbitrap Fusion mass spectrometer in DDA mode. The survey scan was acquired with 60,000 resolution from 800−2000 *m*/*z* with an AGC target of 1 × 10^6^ and a max injection time of 250 ms. The precursors were isolated in the quadrupole within an isolation window of 1.6 *m*/*z*. The top 15 monoisotopic masses with two to four plus charges were selected for fragmentation using stepped HCD (NCE: 33 ± 10). The MS/MS scan was acquired with a 15,000 resolution, an AGC target of 1 × 10^6^, and a max injection time of 250 ms.

### 4.5. Database Searching and Data Analysis

The MS data were searched against the UniProt human proteome database (release 14 January 2020, 20385) using ProteomeDiscoverer 1.4 (PD1.4). The mass tolerance was set at 20 ppm for the precursor ions and 0.03 Da for the fragment ions. Carbamidomethyl (Cys) was chosen as a fixed modification. Oxidation (Met) and deamidated (Asn) were chosen as the variable modifications. Trypsin was chosen for the enzyme, and two missed cleavages were allowed. Peptide-spectrum match (PSM) was validated by the Percolator node. A false discovery rate (FDR) of 1% was estimated and applied as a threshold for identification. The intensity of each peptide was normalized to the total ion current (TIC) and was presented as ion per million (IPM), as previously described [30,49,50]. Protein quantification was achieved based on the abundance of the three most intense peptides from each identified protein [51].

The glycoproteomic data were searched against the UniProt human proteome database using Byonic (Protein Metrics, San Carlos, CA, USA) for the identification of intact *N*-glycopeptides (IGPs). A built-in 182 human glycan database was included in the database search. HexNAc (203.08 Da) was used as a diagnostic ion in the MS/MS spectra. An FDR of 1% was estimated and applied for the validation of the glycopeptide identification. Quantification was achieved with Byologic (Protein Metrics Inc.) using XIC extraction, which uses inputs from both MS1 raw data and Byonic search results to obtain the peptide intensity data. The total abundance of all of the detected *N*-glycopeptides from bovine thyroglobulin, the internal standard, was used to normalize the abundances of the intact *N*-glycopeptides, as previously described [52,53].

### 4.6. Gene Ontology Enrichment Analysis

Gene ontology (GO) enrichment analysis was performed using ClusterProfiler 4.0. Human gene annotation from OrgDb object. A *P. adjust* < 0.05 was considered statistically significant.

## 5. Conclusions

2DG has a broad effect on cellular functions. From a glycoproteomic perspective, for the first time, we investigated the *N*-glycoproteome alterations induced by 2DG using colorectal cancer cells as a model system. Our findings revealed the molecular details underlying the influence of 2DG on protein glycosylation, especially the alterations in the high-mannose types, highlighting some of the important glycosylation events that might be implicated in the anti-tumor effects of 2DG. These data may provide new clues to facilitate novel strategies for cancer treatment by targeting protein glycosylation.

## Figures and Tables

**Figure 1 ijms-23-08251-f001:**
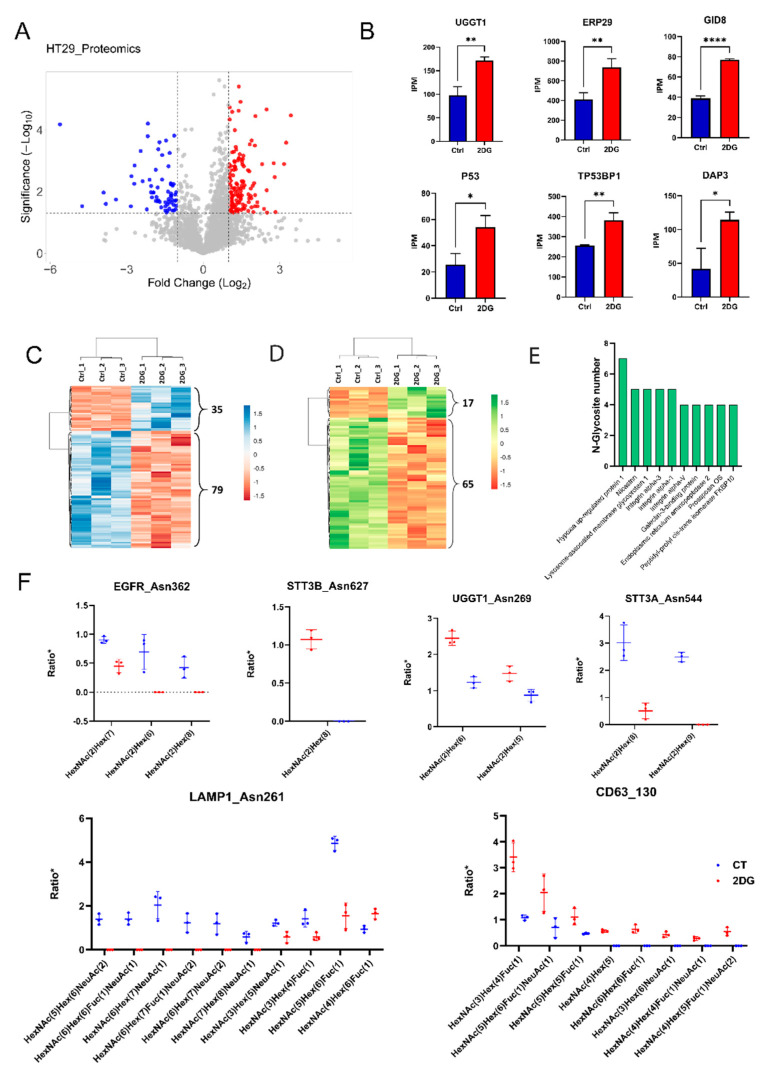
Analysis of the global protein expressional changes and *N*-glycosylation alterations in 2DG-treated HT29 cells. (**A**) Volcano plot displaying the distribution of all proteins with relative protein abundance (2DG/CT fold change ≥ 2 with *p*-value < 0.05, red color; 2DG/CT fold change ≤ 0.5 with *p*-value < 0.05, blue color. *, **, and **** indicates a *p*-value < 0.05, 0.01, and 0.001, respectively). (**B**) Exemplification of six up-regulated proteins associated with 2DG treatment, including UGGT1, UGT1A10, ERP29, TP53, P53I11, and GDI8 (red color represents the relative protein abundance in 2DG-treated HT29 cell line and blue color represents the relative protein abundance in untreated control). (**C**) Hierarchical clustering heatmap of 114 (35 up-regulation and 79 down-regulation) altered IGPs derived from 77 proteins. (**D**) Hierarchical clustering heatmap of 82 (17 up-regulated and 65 down-regulated) altered intact high-mannose *N*-glycopeptides derived from 61 proteins. (**E**) Illustration of *N*-glycoproteins that have the most site-specific *N*-glycoforms. (**F**) Exemplification of the decrease in *N*-glycosylation occupancy at specific glycosites in EGFR (asn362), STT3B (asn627), UGGT1 (asn269), STTA1 (asn544), LAMP1 (asn261), and CD63 (asn130).

**Figure 2 ijms-23-08251-f002:**
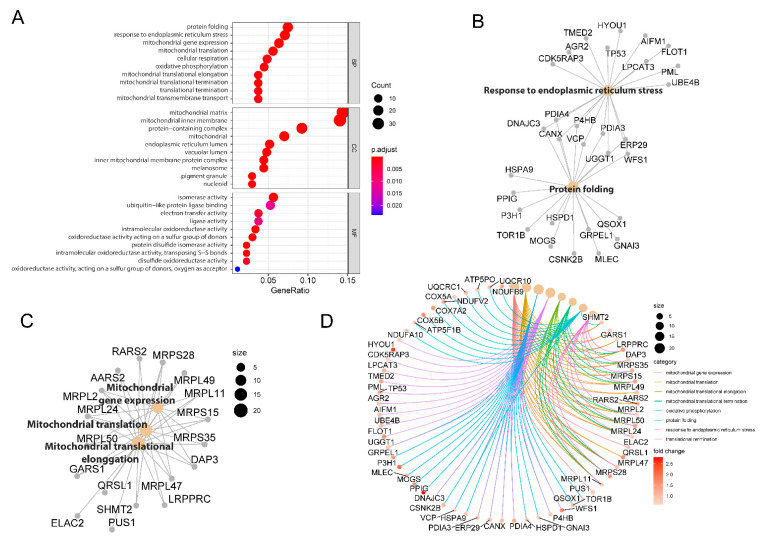
Functional clustering annotation of proteome changes induced by 2DG. (**A**) GO analysis of up-regulated proteins associated with 2DG treatment in the HT29 cell line. Cnetplot plots show the most significant enriched terms with associated proteins. The dotplot diagram was performed in RStudio 4.0 using the ggplot2 R package. (**B**) The cnetplot diagram shows the relationship between the protein sets associated with the biological processes of response to endoplasmic reticulum stress and protein folding. (**C**) The cnetplot diagram shows the relationship among the protein sets associated with biological process of mitochondrial gene expression, mitochondrial translation, and mitochondrial translational elongation. (**D**) The cnetplot-circular diagram shows the 68 up-regulated proteins associated with the eight top-ranked biological processes.

**Figure 3 ijms-23-08251-f003:**
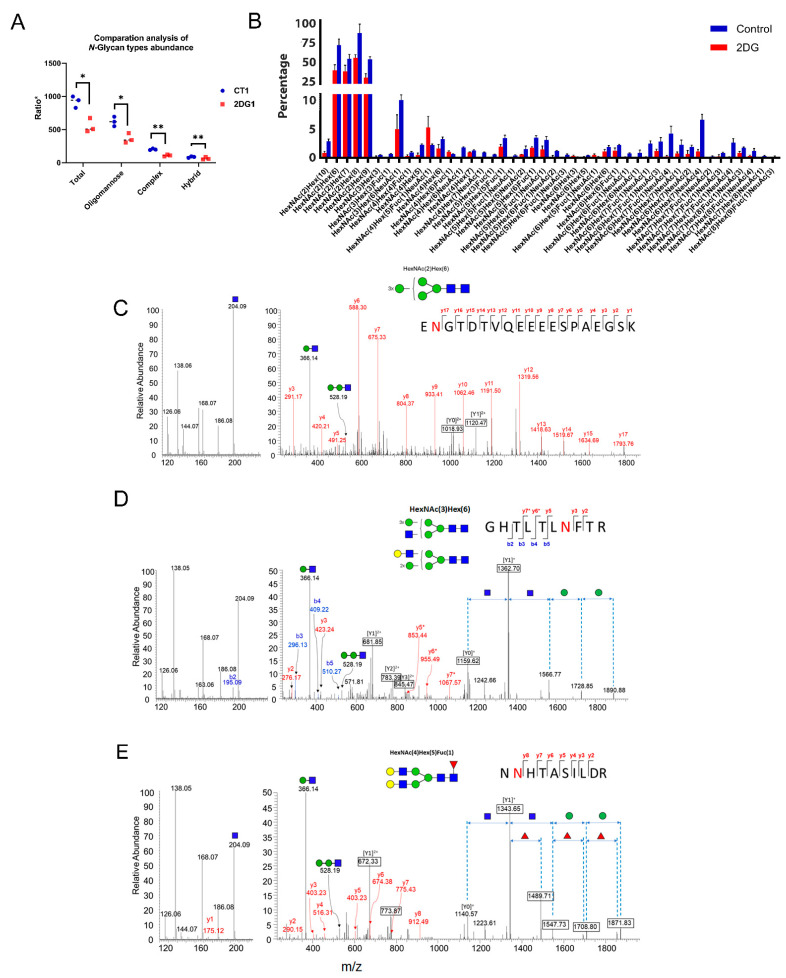
*N*-glycosylation changes induced by 2DG. (**A**) Comparison of the three *N*-glycan types (high-mannose, hybrid, and complex) with and without 2DG treatment. (**B**) Classification of the 1560 IGPs into 42 groups based on their *N*-glycan composition. (**C**–**E**) Three examples representing intact *N*-glycopeptides with high-mannose, hybrid, and complex glycan types, respectively. Note: *, **, indicates a *p*-value < 0.05, 0.01, respectively.

**Figure 4 ijms-23-08251-f004:**
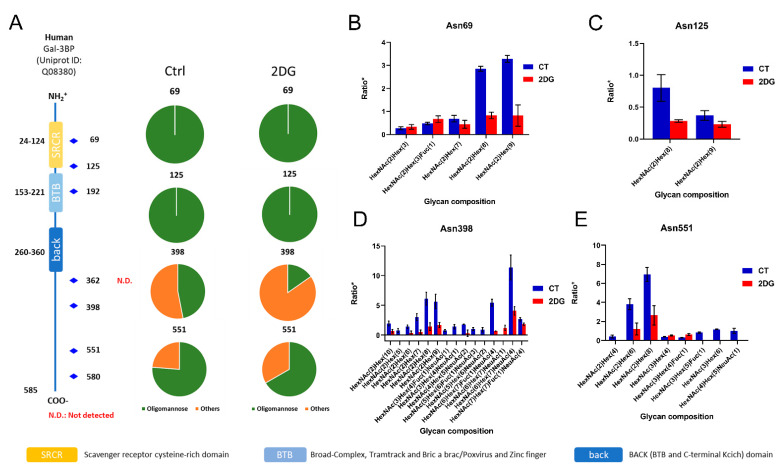
Alterations in the *N*-glycoforms of galectin 3-binding protein in HT29 cells induced by 2DG. (**A**) Site-specific *N*-glycosylation changes in galectin 3-binding protein at *N*-glycosites of Asn69, Asn125, Asn398, and Asn551. (**B**,**C**) Glycoforms with high-mannose identified at Asn69 and Asn125 sites showing a significant decrease or even completely inhibition in glycosylation due to 2DG treatment. (**D**,**E**) Glycoforms at Asn398 and Asn551 with not only high-mannose, but also complex and hybrid *N*-glycans, showing a similar influence from 2DG treatment, in particular mannosylation. Ratio* represents the XIC of the target peptide/internal standard (THYG).

**Figure 5 ijms-23-08251-f005:**
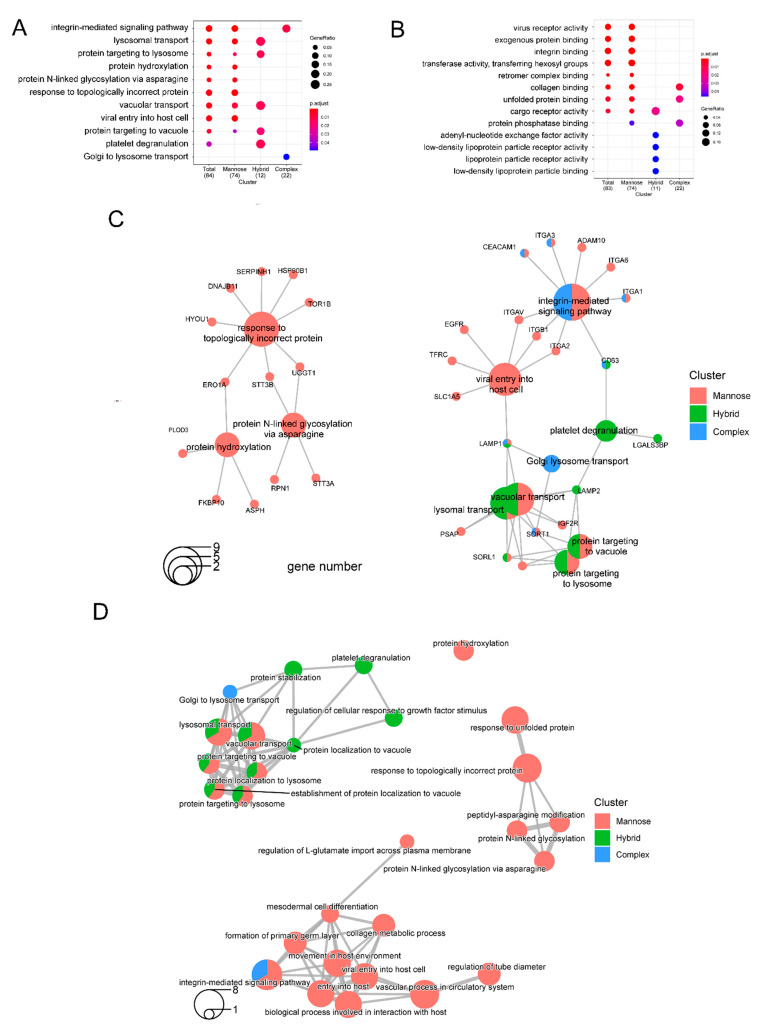
Functional clustering annotation of *N*-glycoproteins with down-regulation of the high-mannose-type induced by 2DG. (**A**) The enriched biological processes associated with the down-regulation of the high-mannose-type, including the integrin−mediated signaling pathway, lysosomal transport, protein targeting to lysosome, protein hydroxylation, and protein *N*-glycosylation via asparagine, etc. (**B**) The corresponding enriched molecular functions, including virus receptor activity, exogenous protein binding, integrin binding, transferase activity, transferring hexosyl group, retromer complex binding, and collagen binding, etc. (**C**) Exemplification of the functional clusters of glycoproteins with down-regulated high-mannose, hybrid, and complex types. (**D**) Exemplification of networks of biological processes involved in glycoproteins with down-regulated *N*-glycosylation with high-mannose, hybrid, and complex types.

**Figure 6 ijms-23-08251-f006:**
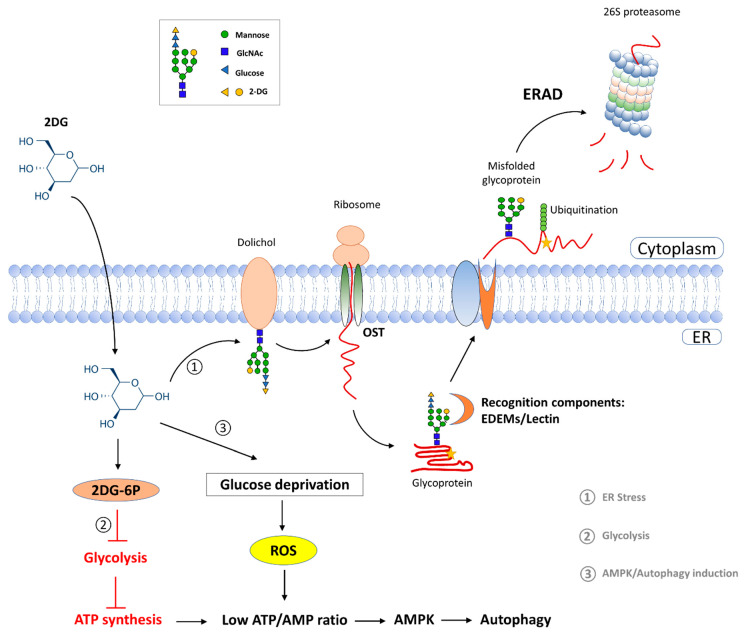
The influence of 2-Deoxy-D-glucose (2DG) on tumor cells. 2DG inhibits the growth and survival of cancer cells by interfering with the ATP produced by the metabolism of D-glucose. In addition, 2DG can inhibit protein glycosylation in vivo by competing with D-mannose, leading to endoplasmic reticulum (ER) stress and misfolded protein responses in cancer cells.

## Data Availability

The data presented in this study are available in the Appendix A. The raw data are available on request from the corresponding authors.

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
