# Peer review of "An Integrated Proteomic and Glycoproteomic Investigation Reveals Alterations in the N-Glycoproteomic Network Induced by 2-Deoxy-D-Glucose in Colorectal Cancer Cells"

_ijms, 2022, doi:10.3390/ijms23158251_

Round 1

Reviewer 1 Report

Ma et al. reported the "Integrated Proteomic and Glycoproteomic Investigation Reveals Alterations in N-Glycoproteomic Network Induced by 2-Deoxy-D-Glucose in Colorectal Cancer Cells". Overall, the manuscript is well written.

The authors addressed the current problems and discussed the possible solution through this study. However, figure 1 is somewhat not necessary in the introduction. authors can move it to the discussion to fit the results of this study to discuss how the associated problems are solved.

Methods are well written and sufficiently elaborated to reproduce the data. 

I can see many citations (31-46) in the results section. As the discussion is in a separate section, no citation is allowed in the results section. Remove these citations.

No supplementary files are available that refer to in the text. Without evaluating them, no comments are appropriate. 

Author Response

Reviewer 1

“Ma et al. reported the "Integrated Proteomic and Glycoproteomic Investigation Reveals Alterations in N-Glycoproteomic Network Induced by 2-Deoxy-D-Glucose in Colorectal Cancer Cells". Overall, the manuscript is well written.”

“The authors addressed the current problems and discussed the possible solution through this study. However, figure 1 is somewhat not necessary in the introduction. authors can move it to the discussion to fit the results of this study to discuss how the associated problems are solved.”

Answer: We have removed the original Figure 1 in the Introduction and placed it in the Discussion section as Figure 6 in the revision.

I can see many citations (31-46) in the results section. As the discussion is in a separate section, no citation is allowed in the results section. Remove these citations.”

Answer: We thank the reviewer for this suggestion. We have removed the citations in the Results section and made changes in the Discussion accordingly.

“No supplementary files are available that refer to in the text. Without evaluating them, no comments are appropriate.”

Answer: We apologize for this neglection. We have submitted the supplementary files with the revised version.

Reviewer 2 Report

Here are my comments about this nice manuscript by Ma et al.:

- Introduction should be more shortened and emphasized on the conducted research that is displayed.

- Material and methods should be more detailed - a lack of clarification is indeed needed for some experiments

- Results part is quite clear - Figure colors shoud be perhaps modified to less "aggressive" ones ?

- Line 415, this citation should be added explaining the role of 2-DG: 

Cassim S, Vučetić M, Ždralević M, Pouyssegur J. Warburg and Beyond: The Power of Mitochondrial Metabolism to Collaborate or Replace Fermentative Glycolysis in Cancer. Cancers (Basel). 2020 Apr 30;12(5):1119. doi: 10.3390/cancers12051119. PMID: 32365833; PMCID: PMC7281550.

Author Response

Reviewer 2

Here are my comments about this nice manuscript by Ma et al.:

- Introduction should be more shortened and emphasized on the conducted research that is displayed.

Answer: We thank the reviewer for the suggestion. We have removed the last part of the first paragraph in the Introduction, i.e. “Studies have shown that activated AMPK could phosphorylate ….”.

- Material and methods should be more detailed - a lack of clarification is indeed needed for some experiments

Answer: We thank the reviewer for the suggestion. We have provided more details in the Material and Methods section “2.3. Enrichment of intact N-glycopeptides” in the revised version.

- Line 415, this citation should be added explaining the role of 2-DG: 

Cassim S, Vučetić M, Ždralević M, Pouyssegur J. Warburg and Beyond: The Power of Mitochondrial Metabolism to Collaborate or Replace Fermentative Glycolysis in Cancer. Cancers (Basel). 2020 Apr 30;12(5):1119. doi: 10.3390/cancers12051119. PMID: 32365833; PMCID: PMC7281550.

Answer: We have added the citation in the revision to explain the relation between 2-DG and Mitochondrial Metabolism, line 310-317.

Reviewer 3 Report

The manuscript presents a very interesting and comprehensive work on the analysis of the effect of 2DG on the proteome and glycoproteome of cancer cells. The study is worthy of attention and discussion by the scientific community. However, for the publication of the article, it is necessary to correct a number of shortcomings made by the authors when assembling the manuscript and analyzing the data. I apologize for not dividing the major and minor points. Here they are all together:

The English language should be slightly edited – for example, see Lines 59, 70, 79, 95, 98, 208, 296, 343, 344, etc.

Line 78 – by components of what?

Line 82 – “The influence of 2DG on tumor cells. 2-Deoxy-D-glucose (2DG) inhibits…” 2DG and 2-Deoxy-D-glucose (2DG) should be swapped with each other. First, you enter the term and give its abbreviation, and only then use the abbreviated version alone.

Line 202 – where is the data sheet file S1? I was unsuccessful to find it.

Lines 201-202 – “A total of 325 up-regulated proteins and 94 down-regulated proteins were identified.” Were all these proteins found in every biological replicate? With different biological repetitions, especially if the time of cell cultivation and the time of their processing with 2DG changed even slightly, some modifications in proteomic patterns can be observed. Please clarify.

The same question is about information represented on Lines 229-230, including the absence of the file S2.

Does the coloring in red and blue have the same meaning in the rest of figure 2 as in case of the A part?

Line 250 – This Figure S1 also cannot be verified because the file with Supplementary materials is missing.

Lines 271-285 – Why did the authors focus only on the analysis of up-regulated proteins, and did not perform similar calculations for down-regulated ones? In my opinion, these data should be included in the manuscript and discussed, they can play an equally important role in changing the metabolic potential of cancer cells.

Line 288 - Here, too, it is necessary to clarify whether the specified 1560 IGPs are the resulting ones, i.e. appearing in all biological repeats under the specified impact?

Lines 294, 397 – The Figures S2 and S3 should be presented among other Figures in the Supplementary material (I didn’t find the last one).

Figure 4B is too small for the adequate interpretation. The same I can say about the whole Figures 5 and 6.

Line 306 – please add “correspondingly”

Author Response

Reviewer 3

The manuscript presents a very interesting and comprehensive work on the analysis of the effect of 2DG on the proteome and glycoproteome of cancer cells. The study is worthy of attention and discussion by the scientific community. However, for the publication of the article, it is necessary to correct a number of shortcomings made by the authors when assembling the manuscript and analyzing the data. I apologize for not dividing the major and minor points. Here they are all together:

The English language should be slightly edited – for example, see Lines 59, 70, 79, 95, 98, 208, 296, 343, 344, etc.

Answer: We thank the reviewer for pointing out the grammatic errors and apologize for the mistakes. We have revised these sentences accordingly.

Line 78 – by components of what?

Answer: We have revised the sentence as following: “Consequently, 2DG modified glycoproteins cause protein misfolding and activation of UPR to maintain cell viability or initiate cell death programs”. Line 63-65.

Line 82 – “The influence of 2DG on tumor cells. 2-Deoxy-D-glucose (2DG) inhibits…” 2DG and 2-Deoxy-D-glucose (2DG) should be swapped with each other. First, you enter the term and give its abbreviation, and only then use the abbreviated version alone.

Answer: We thank the reviewer for pointing this out. We have made the correction accordingly.

Line 202 – where is the data sheet file S1? I was unsuccessful to find it.

Answer: We apologize for the neglection. The supplementary information has now been submitted with the revision.

Lines 201-202 – “A total of 325 up-regulated proteins and 94 down-regulated proteins were identified.” Were all these proteins found in every biological replicate? With different biological repetitions, especially if the time of cell cultivation and the time of their processing with 2DG changed even slightly, some modifications in proteomic patterns can be observed. Please clarify.

Answer: The reviewer’s point is well taken. These differentially expressed proteins were indeed identified in every biological replicate sample. The biological replicates were performed using the same cell culture and 2DG treatment conditions. In addition, in the data analysis, to minimize inter-sample variations, a normalization method, which normalized  the intensity of a peptide to the total ion current (TIC), was applied for label-free quantification. We used fold-change (≥ 2-fold) and t-test (p-value < 0.05) to identify the differentially expressed proteins between 2DG-treated HT29 cells and the untreated controls. The detailed information was provided in the Materials and Methods section. The variations associated with biological replicates were represented by the standard deviation of an average normalized intensity, as exemplified in Figure 1B.

The same question is about information represented on Lines 229-230, including the absence of the file S2.

Answer: Again, we apologize for the neglection. The supplementary information has now been submitted with the revision.

Does the coloring in red and blue have the same meaning in the rest of figure 2 as in case of the A part?

Answer: No, the coloring in red and blue represent different meaning in the plots presented in Figure 1 (original Figure 2), as the plots showed changes of different aspects in protein profiling and N-glycosylation. We have elaborated the color indications for each plot in the legend of Figure 1.

Line 250 – This Figure S1 also cannot be verified because the file with Supplementary materials is missing.

Answer: Again, we apologize for the neglection. The supplementary information has now been submitted with the revision.

Lines 271-285 – Why did the authors focus only on the analysis of up-regulated proteins, and did not perform similar calculations for down-regulated ones? In my opinion, these data should be included in the manuscript and discussed, they can play an equally important role in changing the metabolic potential of cancer cells.

Answer: We agree with the reviewer. In fact, we performed the GO-enrichment analysis for both up-regulated and down-regulated proteins. The results for the up-regulated and down-regulated proteins were presented in Figure 2 and Figure S1, respectively. In contrast to the results of up-regulated proteins, the GO enrichment analysis of down-regulated proteins didn’t find any significant enrichment in biological processes and cellular components in association with 2DG treatment. The enriched cellular functions associated with the down-regulated proteins were presented in Figure S1. We have added the GO analysis result for down-regulated proteins in the revised manuscript (line 191-193).

Line 288 - Here, too, it is necessary to clarify whether the specified 1560 IGPs are the resulting ones, i.e. appearing in all biological repeats under the specified impact?

Answer: In total, we identified 1738 IGPs in the HT29 cell line. We were able to categorize 1560 IGPs into three general types of N-glycans, including high-mannose, complex, and hybrid.  Among these IGPs, 672 were detected in all biological repeats under 2DG-treated or untreated HT29 cell line. This information was provided in line 169-171 and line 211 and 217 in the revision, as well as in Data sheet file S2.

Lines 294, 397 – The Figures S2 and S3 should be presented among other Figures in the Supplementary material (I didn’t find the last one).

Answer: Again, we apologize for the neglection. All supplemental figures have now been submitted with the revision.

Figure 4B is too small for the adequate interpretation. The same I can say about the whole Figures 5 and 6.

Answer: We have resized Figure 4B, as well as Figure 5 and 6 in the revised manuscript.

Line 306 – please add “correspondingly”

Answer: Added “correspondingly” in the revised version.

Round 2

Reviewer 1 Report

The authors revised the manuscript sufficiently. However, they provide the blank data sheet of S1. Additionally, they need to explain figure 6 in the discussion before accepting the paper. 

Author Response

Reviewer 1 (round 2):

The authors revised the manuscript sufficiently. However, they provide the blank data sheet of S1.

Answer: We apologize for our negligence. We have uploaded data sheet S1 in the revised materials.

Additionally, they need to explain figure 6 in the discussion before accepting the paper. 

Answer: We have added additional discussion to explain figure 6 in the Discussion section. Please see line 322-326 in the revised manuscript.

Reviewer 3 Report

I thank the authors for the careful study of all the comments.

Author Response

We thank the reviewer for her/his effort in reviewing this manuscript and the kind comment.